# *Bombyx mori* Pupae Efficiently Produce Recombinant AAV2/HBoV1 Vectors with a *Bombyx mori* Nuclear Polyhedrosis Virus Expression System

**DOI:** 10.3390/v13040704

**Published:** 2021-04-18

**Authors:** Qian Yu, Pengfei Chang, Xiaoxuan Liu, Peng Lü, Qi Tang, Zhongjian Guo, Jianming Qiu, Keping Chen, Qin Yao

**Affiliations:** 1School of Life Sciences, Jiangsu University, Zhenjiang 212013, China; qianyu@ujs.edu.cn (Q.Y.); 15751000531@163.com (P.C.); lxx976163@163.com (X.L.); penglu@ujs.edu.cn (P.L.); tangqi1224@163.com (Q.T.); gzh762677@ujs.edu.cn (Z.G.); 2Department of Microbiology, Molecular Genetics and Immunology, University of Kansas Medical Center, Kansas City, KS 66160, USA; jqiu@kumc.edu

**Keywords:** rBmNPV/AAV2Rep-HBoV1Cap, rBmNPV/AAV2ITR-eGFP, silkworm pupae, BmN cells, rAAV2/HBoV1

## Abstract

Recombinant adeno-associated virus (AAV) vectors have broad application prospects in the field of gene therapy. The establishment of low-cost and large-scale manufacturing is now the general agenda for industry. The baculovirus-insect cell/larva expression system has great potential for these applications due to its scalability and predictable biosafety. To establish a more efficient production system, *Bombyx mori* pupae were used as a new platform and infected with recombinant *Bombyx mori* nuclear polyhedrosis virus (BmNPV). The production of a chimeric recombinant adeno-associated virus (rAAV) serotype 2/human bocavirus type-1 (HBoV1) vector was used to evaluate the efficiency of this new baculovirus expression vector (BEV)–insect expression system. For this purpose, we constructed two recombinant BmNPVs, which were named rBmNPV/AAV2Rep-HBoV1Cap and rBmNPV/AAV2ITR-eGFP. The yields of rAAV2/HBoV1 derived from the rBmNPV/AAV2Rep-HBoV1Cap and rBmNPV/AAV2ITR-eGFP co-infected BmN cells exceeded 2 × 10^4^ vector genomes (VG) per cell. The rBmNPV/AAV2Rep-HBoV1Cap and rBmNPV/AAV2ITR-eGFP can express stably for at least five passages. Significantly, rAAV2/HBoV1 could be efficiently generated from BmNPV-infected silkworm larvae and pupae at average yields of 2.52 × 10^12^ VG/larva and 4.6 × 10^12^ VG/pupa, respectively. However, the vectors produced from the larvae and pupae had a high percentage of empty particles, which suggests that further optimization is required for this platform in the future. Our work shows that silkworm pupae, as an efficient bioreactor, have great potential for application in the production of gene therapy vectors.

## 1. Introduction

The recombinant adeno-associated virus (rAAV) vector has become a safe and efficient general vector in gene therapy. Many rAAV vectors have shown impressive therapeutic effects in various clinical trials [1,2]. rAAV vectors have been used to successfully transduce nondividing cells in muscle, liver, brain, and eye tissue [3,4,5,6]. Although different rAAV serotypes have been characterized for specificity in human airway therapy, they are still unable to produce efficient transduction due to the defense mechanisms and luminal barriers of the cells [7].

The limited packaging ability has also restricted the application of rAAVs in cystic fibrosis (CF) gene therapy [8]. The coding sequence of the large CF transmembrane conductance regulator (CFTR) is 4.43 kb, approaching the 4.679 kb of the AAV small genome size [9]. Human bocavirus type-1 (HBoV1) has a high tropism for the apical membrane of human airway epithelia [10]. The packaging of a rAAV serotype 2 genome into a HBoV1 capsid produces a chimeric vector (rAAV2/HBoV1) that also efficiently transduces human airway epithelia [11]. This chimeric vector has shown potential for use in gene therapies for cystic fibrosis (CF) and other lung diseases [9,10,11].

The baculovirus expression vector system (BEVS)—a multifunctional platform for the expression of either individual recombinant protein or multimeric protein complexes, virus-like particles, and membrane proteins—has been widely used for different research purposes [12]. Recently, *Autographa californica* nuclear polyhedrosis virus (AcMNPV) has demonstrated particular utility in the commercial-scale manufacture of vaccines and gene therapy vectors [13,14]. Besides, *Bombyx mori* nuclear polyhedrosis virus (BmNPV) has shown a valuable potential for the expression of several proteins. Due to the host specificities, the recombinant AcMNPV can only express foreign genes in *Spodoptera frugiperda* and *Tricoplusia ni* cell lines or larvae; the recombinant BmNPV can express a genome of interest in either the *Bombyx mori* cell line (BmN) or *Bombyx mori* larvae and pupae, respectively. The protein expression efficiency when using *Bombyx mori* larvae and pupae has been reported to be 10- to 100-fold higher than that using BmN cells [14]. Since the successful generation of α-interferon was reported in *Bombyx mori* larvae in 1985 [15], a variety of recombinant proteins, mostly based on the BmNPV infection of the silkworm, have been produced via this platform [16,17,18,19].

The baculovirus-mediated production of rAAV vectors in insect cells is particularly well suited for the production of large quantities of rAAV [20]. A variety of flexible insect cell baculovirus production systems have been developed for rAAV production to date [21,22,23]. With the improvement of gene packaging structure and system complexity, the yield of rAAV has become increasingly high. Compared to the traditional mammalian cell production system, the BEVS has many advantages, such as (1) strong polyhedrin or p10 promoters that can express foreign proteins at high levels, (2) the fact that insect cells can be cultured without serum suspension at low cost and are easy to scale, and (3) the fact that baculovirus is not pathogenic to the human body and that the downstream purification of rAAV vectors is easy [24,25].

The silkworm (*Bombyx mori*), as a bioreactor and an important economic insect in China, has also proved to have great potential for the large-scale production of eukaryotic proteins as well as recombinant baculoviruses for gene transfer to mammalian cells. In this study, we report a high-level rAAV2/HBoV1 vector production platform. We demonstrated that the chimeric rAAV2/HBoV1 vector can be efficiently produced in silkworm larvae, pupae, and their cell line (BmN) by the direct injection or infection with the recombinant baculoviruses. Silkworm pupae showed the highest production efficiency, with an average rAAV2/HBoV1 yield exceeding 4.6 × 10^12^ VG per pupa. The silkworm, as an important model organism, could also be a powerful tool for the large-scale production of different rAAV vectors.

## 2. Materials and Methods

### 2.1. Plasmid and Recombinant Baculovirus Expression Vector (BEV) Construction

The two recombinant plasmids used in the project were constructed into pFastBacDual cloning vector (Invitrogen, Carlsbad, CA, USA) which was described in the previous work [26] and specific plasmids construction information was provided in the Appendix A.

Following the manufacturer’s instructions for the Bac-to-Bac Baculovirus Expression System (Invitrogen), we constructed two Bacmids by the transposition of the above two plasmids with DH10BmBac^TM^
*Escherichia coli* competent cells. Bacmid DNA isolated from amplified bacterial colonies was used to transfect BmN cells to generate recombinant *Bombyx mori* nuclear polyhedroviruses (BmNPV), which were named rBmNPV/AAV2Rep-HBoV1Cap and rBmNPV/AAV2ITR-eGFP. Serial passages of rBmNPV/AAV2ITR-eGFP were then titrated according to the median tissue culture infectious dose (TCID_50_), and rBmNPV/AAV2Rep-HBoV1Cap was titrated in plaque forming units (pfu) by a plaque assay as described in the manual of the Bac-to-Bac Baculovirus Expression System (Invitrogen)^®^ (Figure 1).

### 2.2. Cell Culture and Silk Worm Rearing

BmN cells were cultured in TC-100 supplemented with 10% fetal bovine serum (FBS) (Gibco, Thornton, NSW, Australia) and 1% Penicillin-Streptomycin (Gibco, Grand Island, NY, USA), maintained in a 27 °C incubator. The *Bombyx mori* larvae were fed with mulberry leaves at 25 °C and 60% humidity up to the fifth instar and injected with virus. After 5–6 days of the cocooning of *Bombyx mori* larvae, we cut the cocoons open and took out the pupae. Similar to *Bombyx mori* larvae, after the injection of recombinant viruses, the pupae were cultured in the dark at 25 °C and 60% humidity for 3–4 days.

Human normal bronchial epithelial (16HBE) cells were cultured in Dulbecco’s Modified Eagle Medium (DMEM) (Beijing, China) supplemented with 10% fetal bovine serum (FBS) (Gibco, Thornton, NSW, Australia) at 37 °C and 5% CO_2_.

### 2.3. Larva or Pupa Injection and Observation 

As previously described when the *Bombyx mori* larvae reached the fifth instar stage, they were injected with 5 uL (~1 × 10^6^ VG) each of two the recombinant baculoviruses (or sterile PBS buffer as uninfected control) near the proleg (forward from the body cavity) [15]. For the observation of pathological symptoms after the injection, the larvae were narcotized on ice, and low-magnification imaging was carried out by stereomicroscope (Olympus SZX16, Shinjuku-ku, Tokyo, Japan) to obtain the whole-larvae fluorescence images. Infected larvae were washed three times with PBS buffer (pH 7.4, Gibco, Grand Island, NY, USA) to collect and stored at −80 °C.

Like the larvae, pupae were injected with 5 µL (~1 × 10^6^ VG) each of two recombinant baculoviruses (or sterile PBS buffer as uninfected control). For the observation of pathological symptoms after the injection, low-magnification imaging was carried out by stereomicroscope (Olympus SZX16, Shinjuku-ku, Tokyo, Japan) to obtain the whole-pupae fluorescence image. Infected pupae were stored at −80 °C to use.

### 2.4. rAAV2/HBoV1 Production and Purification

For adherent cultured BmN cells, we prepared BmN cells in a six-well format (ThermoFisher, Suzhou, Jiangsu, China) at 80% confluence and used the Bac-to-Bac protocol to transfect two recombinant bacmid DNA constructs using Cellfectin^®^ II Reagent (Gibco, Van Allen Way, Carlsbad, CA, Lithuania). 3–4 days of transfection, the supernatant was collected, and we obtained the two recombinant BmNPV P1 stocks. The P1 viral stock was a small-scale, low-titer stock, and we used this stock to infect cells to generate a high-titer P2 stock or other higher-titer stocks.

We used the TCID_50_ and plaque forming unit (pfu) assay, finding that the P3 viral stock was the highest-titer BmNPV. Next, we used two BmNPV P3 stocks (~1 × 10^12^ pfu/mL) to co-infect BmN cells to produce the rAAV2/HBoV1. At 72 h post-infection, the cells were collected by centrifugation and lysed in PBS (pH 7.4, Gibco, Grand Island, NY, USA) for purification. Detailed virus purification procedures were provided in Appendix A.

For the *Bombyx mori* larvae or pupae, they were infected by co-injecting two recombinant BmNPVs and collected to prepare for the follow-up experiments. The larvae or pupae were ground into powdered samples with liquid nitrogen and dissolved in PBS (pH 7.4, Gibco, Grand Island, NY, USA) for purification. The subsequent experiments were the same as the purification in BmN cells.

### 2.5. SDS-PAGE and Western Blotting

Proteins were separated by 10% sodium dodecyl sulfate-polyacrylamide gel electrophoresis (SDS-PAGE). After dyeing with Coomassie brilliant blue R250 dye for 1 h, the gel was decolorized overnight on a shaker with a decolorizing solution. For the Western blot analysis, the proteins in the polyacrylamide gels were transferred to nitrocellulose membranes (Immolon^®^-P, Tullagreen, Carrigtwohill, Co Cork, IRL). Rep proteins were detected with an anti-AAV2 Rep monoclonal antibody (Fitzgerald, USA) at a 1:100 dilution. Cap proteins were detected with an anti-HBoV1 VP2 monoclonal antibody at a 1:3000 dilution (Zoombio Biotechnology, Nanjing, Jiangsu, China). A monoclonal antibody against β-tubulin (Proteintech, Wuhan, Hubei, China) was used at a 1:2000 dilution to normalize the added protein content. A monoclonal antibody against eeGFP (Transgen, Beijing, China) was used at a 1:2000 dilution. Secondary detection antibodies (Transgen, Beijing, China) were used at 1:2000 dilutions. The blots were imaged using Enhanced Chemiluminescence (Tanon, Shanghai, China) solutions with the Tanon chemiluminescence imaging system.

### 2.6. Quantification of rAAV2/HBoV1

The rAAV2/HBoV1 titers were determined by a quantitative PCR (qPCR) assay with the ChamQ Universal SYBR Green Master Mix (Vazyme, Nanjing, China). A plasmid containing a eGFP ORF was used to establish a standard curve for absolute quantification. Quantitative PCR analysis was performed with a primary denaturation step of 30 s at 95 °C, followed by 40 cycles of 95 °C for 10 s and 60 °C for 30 s, on a 7300 Fast Real-Time PCR System (Applied Biosystems™ QuantStudio™ 3, USA) using the ChamQ Universal SYBR Green Master Mix reagent. The sequences of the primers specific to the eGFP sequence were showed in the Appendix A.

### 2.7. Transmission Electron Microscopy

The virus particles were assessed by negative stain and transmission electron microscopy. Briefly, approximately 3–5 µL samples were spotted onto a 400-mesh carbon-coated copper grid, incubated for 1 min, quickly removed and air-dried. Then the grid was stained with 2% phosphotungstic acid (pH 7.4) for 2 min, removed immediately by adsorption to filter paper, washed with a drop of water, and air-dried again. Grids were examined using a JEM2100 (HR) transmission electron microscope (JEOL, Tokyo, Japan) at a magnification setting of 30,000× and an accelerating voltage of 100 kV.

## 3. Results

### 3.1. Production of Infective Recombinant BmNPV in Cultured BmN Cells

In this study, we used the *Bombyx mori* nuclear polyhedrosis baculovirus (BmNPV) expression system, in which the rAAV2/HBoV1 vectors were produced by co-infecting BmN cells or silkworm pupae with two BmNPV recombinant baculoviruses carrying different essential AAV genes. First, we followed the introduction of Bac-to-Bac system to construct *E.coli*BmDH10Bac/AAV2Rep-HBoV1Cap and *E.coli*BmDH10Bac/AAV2ITR-eGFP. The recombinant bacmids were identified by PCR with primers specific for AAV2 ITR, AAV2 Rep, and HBoV1 Cap as well as the Bac-to-Bac system general primer M13F/R.

After obtaining positive results, the two recombinant bacmids were transfected into BmN cells with Cellfectin reagent II (Invitrogen, Carlsbad, CA, USA). By 72 h post-transfection, the BmN cells transfected with the AAV ITR/eGFP bacmid showed apparent eGFP expression, and the BmN cells transfected with the AAV2Rep/HBoV1Cap bacmid showed clear cytopathic effects. We then performed Western blotting verification successfully with antibodies against eGFP, AAV2Rep, and HBoV1 Cap for BmN cell pellets. The cell-cultured suspensions were collected as the P1 stock of the required recombinant rBmNPV/AAV2Rep-HBoV1Cap and rBmNPV/AAV2ITR-eGFP (Figure 1).

According to the instructions for Bac-to-Bac, the P3 stock of baculovirus had the highest titer; therefore, we infected BmN cells with the obtained P1 stock to produce the P2 and P3 stocks. rBmNPV/AAV2ITR-eGFP was titrated according to the median tissue culture infectious dose (TCID_50_), and the titer of the P3 stock was approximately 6.9 × 10^12^ pfu/mL. rBmNPV/AAV2Rep-HBoV1Cap was titrated in pfu by a plaque assay, and the titer of the P3 stock was about 2.5 × 10^12^ pfu/mL.

### 3.2. Stability of rBmNPV/AAV2Rep-HBoV1Cap and rBmNPV/AAV2ITR-eGFP upon Serial Passage

One of the major challenges faced when using the baculovirus expression system is that the existence of multiple viruses in this system and the integration of multiple gene expression elements in the recombinant baculoviruses leads to genetic instability in the AAV2/HBoV1 vector production process; the stable expression of AAV-related genes is essential for the large-scale production of AAV vectors.

To address this, we amplified the rBmNPV/AAV2Rep-HBoV1Cap and rBmNPV/AAV2ITR-eGFP for serial amplification passages from Passage 1 to Passage 8. For each passage, the BmN cells were infected with recombinant baculoviruses at a multiplicity of infection (MOI) of 0.5, and the cell-cultured supernatants were harvested as rBmNPV/AAV2Rep-HBoV1Cap and rBmNPV/AAV2ITR-eGFP stocks after 72 h post-infection. Then, the BmN cells cultured in a 12-well plate at 80% confluence were co-infected with rBmNPV/AAV2Rep-HBoV1Cap and rBmNPV/AAV2ITR-eGFP at an MOI of 0.5 (total MOI of 1) from Passage 1 to Passage 8.

After 96 h of co-infection, we observed the eGFP expression of the infected BmN cells and captured images with an Olympus IX73 fluorescence microscope (Figure 2A). The expression of eGFP increased from the P1 to P4 stock, and decreased significantly from the P6 stock. Cytopathic effects were also seen under the microscope under a bright field. To determine whether AAV2 Rep and HBoV1 Cap were expressed successfully, the passage cell samples were analyzed by Western blotting with antibodies against the eGFP, AAV2optRep78/52, and HBoV1 VP2 proteins (Figure 2B).

The expression of the optRep78/Rep52 and HBoV1 VP2 proteins was observed. Stable expression of the AAV2Rep and HBoV1Cap proteins was observed from Passage 1 to Passage 4. Both Rep and Cap expression began to decrease from Passage 6. The Western blotting results were consistent with those of the fluorescence microscopy. These results demonstrate that the stability of the rBmNPV/AAV2Rep-HBoV1Cap and rBmNPV/AAV2ITR-eGFP was maintained over at least four serial passages, and they were stable enough to support sufficient baculovirus stock amplification for large-scale rAAV2/HBoV1 vector production.

### 3.3. Production of rAAV2/HBoV1 in Silkworm Larvae and Pupae by Direct rBmNPV/AAV2Rep-HBoV1Cap and rBmNPV/AAV2ITR-eGFP Infection

In addition to insect cells, insect larvae or pupae have been widely used as a cost-efficient bioreactor for recombinant protein expression, and the protein expression is 10- to 100-fold higher than that achieved with insect cells. The host requirements for the baculovirus are very strict; for instance, the AcNPV can only infect *Spodoptera frugiperda* and *Tricoplusia ni*; however, the BmNPV can only infect *Bombyx mori* larvae or pupae and their cell lines. Therefore, we sought to test whether rAAV2/HBoV1 vector could be generated from *Bombyx mori* larvae and pupae upon co-infection with the novel rBmNPV/AAV2Rep-HBoV1Cap and rBmNPV/AAV2ITR-eGFP.

To test this, fifth-instar *Bombyx mori* larvae were directly injected with the recombinant rBmNPV/AAV2Rep-HBoV1Cap and rBmNPV/AAV2ITR-eGFP. At 3–4 days post-infection, the infected larvae showed apparent pathological symptoms, which were not observed in uninfected larvae, and the infected larvae were smaller than the uninfected larvae (Figure 3A). We also observed remarkable eGFP expression throughout the entire bodies of the infected larvae with a stereomicroscope (Olympus SZX16, Shinjuku-ku, Tokyo, Japan) but not in the uninfected larvae (Figure 3B).

To further analyze which part of the silkworm could produce rAAV2/HBoV1 the most efficiently, we dissected the infected silkworm larvae and obtained six tissues: the cuticula (1), silk gland (2), midgut (3), cephalic region (4), fat body (5), and hemolymph (6) (Figure 3C). EGFP expression was the strongest in the fat body and hemolymph. The results indicate that the larvae can be efficiently infected with the novel rBmNPV/AAV2Rep-HBoV1Cap and rBmNPV/AAV2ITR-eGFP. We tested the AAV2Rep and HBoV1 Cap expression in larva or tissue samples and found the same positive results.

Viral proteins are typically efficiently synthesized in silkworm pupae [19]. Therefore, similarly to what was performed for the larvae, we used direct injection to co-infect silkworm pupae after cocooning for 5 to 6 days with rBmNPV/AAV2Rep-HBoV1Cap and rBmNPV/AAV2ITR-eGFP. At 48 h post-infection, the pupae began to express eGFP and exhibited sufficient fluorescence at 72 h (Figure 3D).

We verified the expression of Rep and Cap by Western blotting, and the results were also in line with our expectations. The expression of eGFP was faster and more efficient in the pupae than the larvae. The western blotting of infected larvae and pupae was showed in the Appendix A. Previous research has shown that the amount of exogenous protein expressed in five silkworm pupae is equivalent to that expressed in 1 L of insect cells [27]. The success of this experiment indicates that silkworm pupae are a potential platform for AAV vector production.

### 3.4. Characterization of rAAV2/HBoV1 Derived from Silkworm Pupae

After the successful infection experiment for the silkworm pupae, we co-infected a large number of silkworm pupae with rBmNPV/AAV2Rep-HBoV1Cap and rBmNPV/AAV2ITR-eGFP to prepare rAAV2/HBoV1 production samples. To characterize the rAAV2/HBoV1 produced by the rBmNPV/AAV2Rep-HBoV1Cap and rBmNPV/AAV2ITR-eGFP, we used iodixanol gradient ultracentrifugation to purify the pupa samples with an SW41Ti rotor in an Optima™XPN ultracentrifuge (Beckman Coulter, Kraemer Blvd. Brea, CA, USA). We compared this method with traditional CsCl density gradient ultracentrifugation. After validation by SDS-PAGE and Western blotting, the rAAV2/HBoV1 was further purified by secondary iodixanol gradient ultracentrifugation.

The crude samples were repeatedly concentrated by ultrafiltration with an Amicon^®^ Ultra 100K device to further improve the purity of the rAAV2/HBoV1. Transmission electron microscopy (TEM) (JEOL, Tokyo, Japan) data showed that the virions of rAAV2/HBoV1 showed the typical icosahedral morphology of a parvovirus particle: relatively uniform spherical particles with a diameter of 20–26 nm (Figure 4A). The electron micrograph showed that the proportion of empty particles was quite high, which is a problem we will need to solve later. A study has implied that this could be related to the lack of the HBoV1 NP1 protein, which is important for HBoV1 capsid packaging [26].

To compare their rAAV2/HBoV1 productivity, we used the same method to purify BmN cells and larvae by iodixanol gradient ultracentrifugation and quantitatively analyzed the production yield differences of three samples by quantitative PCR (qPCR) (Figure 4B). When the BmN cell density was almost 1 × 10^6^ cell/mL, the yield of BmN cells was 2.09 × 10^10^ VG/mL. For the larvae and pupae, each larva or pupa was prepared in 1 mL of PBS for purification. As expected, the rAAV2/HBoV1 yield of the silkworm larvae was lower than that of the pupae. The data showed that the average yield of vectors per larva was 2.52 × 10^12^ VG, and the yield per pupa was 4.60 × 10^12^ VG. These results suggest that silkworm larvae and pupae are both cost-efficient rAAV2/HBoV1 vector production platforms from the perspective of the vector yield.

Since the rAAV2/HBoV1 in this study was a chimeric virus, it was necessary to verify its transduction activity. We infected a human normal bronchial epithelial cell line (16HBE) with rAAV2/HBoV1 derived from pupae at an MOI of ~2000 DRP/cell. At 7 days post-infection, the cells began to show eGFP expression under a fluorescence microscope, and we observed the expression of eGFP until 10 days (Figure 4C). The results demonstrated that the rAAV2/HBoV1 derived from the new *Bombyx mori* baculovirus expression system exhibited transduction activity.

## 4. Discussions

In the past two decades, virus-like particles, such as AAV2 [25], parvovirus B19 [28], and avian AAV [29], have been successfully synthesized using the BEV expression system. rAAV vector production in Sf9 cells was first reported in 2002 using a ThreeBac system [25], which spurred new excitement in the field of manufacturing. Then, the improved TwoBac, OneBac, and MonoBac systems were established and exhibited advanced functionality and practicability [30].

Among the scalable rAAV production systems developed, the insect cell/baculovirus expression system is the most promising platform for the large-scale production of rAAV vectors [31]. The infection with recombinant BEVs of insect cell suspension is more convenient than using the three-plasmid transfection system for HEK293 cells and the downstream production is also simpler than that with mammalian cells for large-scale applications.

Therefore, BEVs and insect cells have good potential for large preparations using a Bio-Reactor, such as those for silkworms and other economic insects. The most significant milestone for the utilization of the BEV system was the breakthrough designation by the US FDA of BioMarin’s Hemophilia A gene therapy candidate rAAV5 vectors, which were produced in insect cells via BEVs [32]. However, the phase III clinical trial of this therapy has been turned down by FDA recently, which suggested a more in-depth characterization of the rAAV vector safety and potency produced from this platform.

Large-scale and low-cost vector production has always been a problem in the application of recombinant viral vectors for gene therapy. Although many different rAAV serotypes have been successfully produced in Sf9 cells and beet armyworm larvae [33] by rAcMNPV infection, there are, to date, no reports regarding other insect production platforms. The *Bombyx mori* silkworm has been reported as an alternative host for the efficient large-scale production of recombinant proteins.

Several proteins, such as α-interferon [15], the green fluorescent protein [16], and the HBeAg protein [17], have been successfully expressed in *Bombyx mori* cell lines and larvae/pupae using the BmNPV baculovirus vector. All of this work demonstrates the great potential of silkworm larvae and pupae for the large-scale expression of foreign genes and even for the preparation of recombinant virus particles.

The rAAV2/HBoV1 vector, which we chose to generate via a BmNPV/pupa platform, has a high tropism for human airway epithelia and is able to encapsidate an oversized rAAV2 genome of 5.8 kb, representing one of the best rAAV vectors for gene delivery to the human airways and holding much promise for use in preclinical trials of CF gene therapy in ferrets and human trials involving CF patients [34]. To establish an efficient production platform for either original rAAV or chimeric rAAV vectors, in this study, we used two baculovirus transfer vectors (pFastBacAAV2ITR-eGFP and pFastBac-AAV2Rep-HBoV1Cap) including the elements necessary for rAAV2/HBoV1 construction, and generated the shuttle bacmids via its BmDH10Bac^TM^
*E.coli* transformant. Two recombinant BmNPVs that integrated all the packaging elements for rAAV2/HBoV1 production were collected from the BmN cells after bacmid transfection using Cellfectin. The stability of the expression of the eGFP, Rep, and Cap proteins persisted from Passages 3 to 5 and decreased from Passages 6 to 8. The decreased expression could have been due to the loss of the ITR-transgene cassette during amplification [26], or caused by competition between the recombinant NPV and contaminant wild NPV.

An extra purification of recombinant BmNPV may be necessary for the further improvement of the stability. Both the *Bombyx mori* larvae and pupae maintained constantly high-level protein expression after the co-infection with rBmNPV/AAV2Rep-HBoV1Cap and rBmNPV/AAV2ITR-eGFP. We found that the rBmNPV/AAV2Rep-HBoV1Cap and rBmNPV/AAV2ITR-eGFP replicated efficiently throughout the larva and pupa bodies ~72 h after the injections. Almost on the sixth day, the silkworm larvae began to die from BmNPV, and the pupae also appeared blackened. Similarly, to the secreted protein expressed by the BmNPV/larvae system, the fat body and hemolymph showed the highest levels among the different tissues [14].

In addition to the direct injection of rBmNPV/AAV2Rep-HBoV1Cap and rBmNPV/AAV2ITR-eGFP, we also examined the protein expression in *Bombyx mori* pupae following the injection of the mixture of recombinant bacmid DNA and DEAE reagent. The pupae injected with the bacmid DNA showed lower protein expression and took a longer time to express the gene of interest (data not shown). The average yield of rAAV2/HBoV1 purified from BmN cells was determined to be 2 × 10^4^ VG/cell, lower than that for the previously reported Sf9 cell preparation. This lower yield is due to the fact that the BmN cell line is currently unable to be cultured in suspension. We also found that rAAV2/HBoV1 is not only produced in BmN cells but also partially released in the cell culture medium. This is similar to the production of rAAV in Sf9 cells, where studies have shown that approximately 30% of the rAAV is released in cell medium. In using recombinant BmNPV produced by the BmN cell line to infect silkworm pupae, we achieved extremely high yields of the vector.

The average yield of vectors per larva pupa was determined to be 2.5 × 10^12^ VG and per pupa was 4.6 × 10^12^ VG. This productive efficiency for each pupa is equal to that for 1 L of Sf9 suspended cell preparation and 2 L of BmN cell preparation. However, even with the pupae that achieved the highest production quantity, we noticed that our preparation still had a high level of empty particles. The rAAV2 genomes may not be as well packaged in HBoV1 capsids as those in AAV2 capsids via the BEV/insect system.

Deng et al. reported that, with the co-infection of a BEV expressing HBoV1 NP1, the rAAV2/HBoV1 vector was produced in a higher quantity and with a lower percentage of empty particles [26]. The expression of NP1 led to an increase in rAAV2 replicative-form (RF) DNA intermediates, which may be responsible for the enhanced production [11]. Another vital reason may be that the ratio of the AAV2 genome and the capsid gene of HBoV1 was not optimal. The low transduction activity of purified rAAV2/HBoV1 observed in 16HBE cells may be due to monolayer culture that is appropriate for transduction. Polarized human airway epithelium cultured at an air-liquid interface (HAE-ALI) is the best cell culture model for wild type HBoV1 infection as well as rAAV2/HBoV1 vector transduction in vitro [11]. We are attempting to establish the HAE-ALI culture model for the study of this chimeric vector in the future.

The purification method of rAAV has a great influence on the yield and quality of the vector. Cesium chloride (CsCl) density gradient ultracentrifugation is one of the most mature and widely used methods for the purification of rAAV in the laboratory, as it applies to all rAAV serotypes and is especially suitable for the removal of empty particles. Here, we also compared the CsCl method with iodixanol discontinuous density gradient ultracentrifugation for the purification of rAAV2/HBoV1 from silkworm pupae. The vectors purified by the CsCl method have much lower purity and full particle percentages. Due to the higher density of cesium chloride, the separation time for rAAV2/HBoV1 in the CsCl medium is much longer than that in the iodixanol medium. Generally, a round of CsCl density gradient centrifugation requires at least 30 h, while the iodixanol method, from our experience, only needs 3–5 h. In addition, according to our repeated experiments, the purity of rAAV2/HBoV1 following iodixanol density gradient centrifugation was much higher than that following CsCl density gradient centrifugation; therefore, multiple rounds of CsCl purification were required to achieve a purity similar to that realized with iodixanol discontinuous density gradient ultracentrifugation. It is worth noting that although density gradient ultracentrifugation has many advantages, its scalability in producing rAAV vectors has always been a major challenge.

Most of the large-scale production of rAAV still relies on the mammalian cell transfection system under the current standard of Good Manufacturing Practice (GMP). AcMNPV/Sf9 cells remain the only insect production platform studied for rAAV manufacture. However, this insect cell–baculovirus expression vector system exhibits altered capsid compositions and lower biological potencies as major drawbacks [35,36]. Most of the AAV serotypes produced in the BEV/insect cell system are characterized by low transduction efficiencies compared with HEK293-derived vectors due to a suboptimal content of VP1 capsid proteins and phospholipase A2 activity. Wu et al. reported that rAAV2 could be efficiently generated from BEV-infected beet armyworm larvae at a per-larva yield of 2.75 ± 1.66 × 10^10^ VG and optimized the system through the establishment of a stable Rep/Cap expression insect cell line [33,37]. Kondratov et al. reported a new insect cell-based production platform utilizing an attenuated Kozak sequence and leaky ribosome scanning to achieve a serotype-specific modulation of AAV capsid protein stoichiometry, which can provide a scalable platform overcoming the main drawbacks [36]. In our case, for the cross-genus pseudo-package of recombinant rAAV2/HBoV1 vectors, there are more considerations for future improvements, such as the incorporation of the essential genes from both HBoV1 and AAV2 or looking for other helper components to solve the package problem.

In conclusion, we established a new BEV/insect expression system for rAAV2/HBoV1 vector production in silkworm pupae infected with recombinant BmNPVs. This new system could successfully pseudo-package the rAAV2 genome in the HBoV1 capsid in silkworm pupae. The yield of the rAAV2/HBoV1 vector in silkworm pupae is higher than that in BmN cells and silkworm larvae and is much higher than that produced by the AcMNPV/Sf9 system. However, the vector production from this new *Bombyx mori* pupa expression platform still has disadvantages compared to the counterpart rAAV2 vector system, such as a high percentage of empty particles and low transduction activity. However, this work provides a hint for the application potential of silkworm insect bioreactors. Our work in the future will focus on the optimization of this production platform for either rAAV or chimeric rAAV vectors.

## Figures and Tables

**Figure 1 viruses-13-00704-f001:**
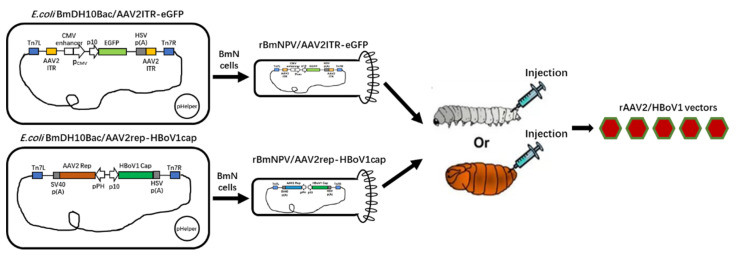
Scheme of the recombinant adeno-associated virus (rAAV)2/human bocavirus type-1 (HBoV1) vectors produced in *Bombyx mori* larvae and pupae upon infection by the recombinant *Bombyx mori* nuclear polyhedrosis virus (BmNPV) bacmid system. The functional elements are represented by open boxes and separated by vertical lines. The promoters are represented by arrows. pHelper: The mini-Tn7 element on the pFastBac plasmid can transpose to the mini-attTn7 target site on the bacmid to construct the recombinant bacmid in the presence of transposition proteins provided by the helper plasmid (Bac-to-Bac manual, Invitrogen^®^).

**Figure 2 viruses-13-00704-f002:**
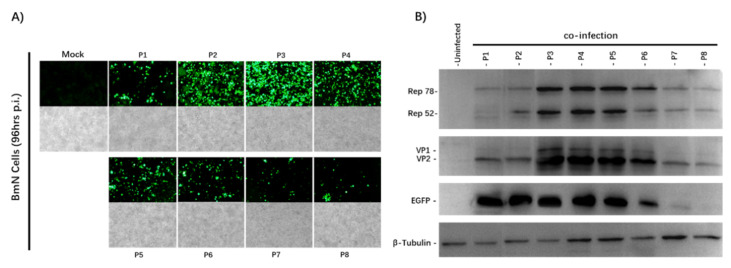
Analysis of the rBmNPV/AAV2Rep-HBoV1Cap and rBmNPV/AAV2ITR-eGFP stability upon serial passages. (**A**) Fluorescence microscopy observations of 6-well plates cultured with BmN cells co-infected with rBmNPV/AAV2ITR-eGFP and rBmNPV/AAV2Rep-HBoV1Cap from P1 to P8 stocks at an MOI of 0.5 (total MOI of 1). The images of BmN cells were taken at 96 h post-infection with an Olympus IX73 fluorescence microscope. (**B**) Western blot analysis of the Rep and Cap protein expression. The passage number is indicated above each lane. The AAV2 Rep, HBoV1 Cap, and eGFP proteins and endogenous cellular protein β-tubulin are indicated in the left margin. The proteins of BmN cells were taken at 96 h post-co-infection.

**Figure 3 viruses-13-00704-f003:**
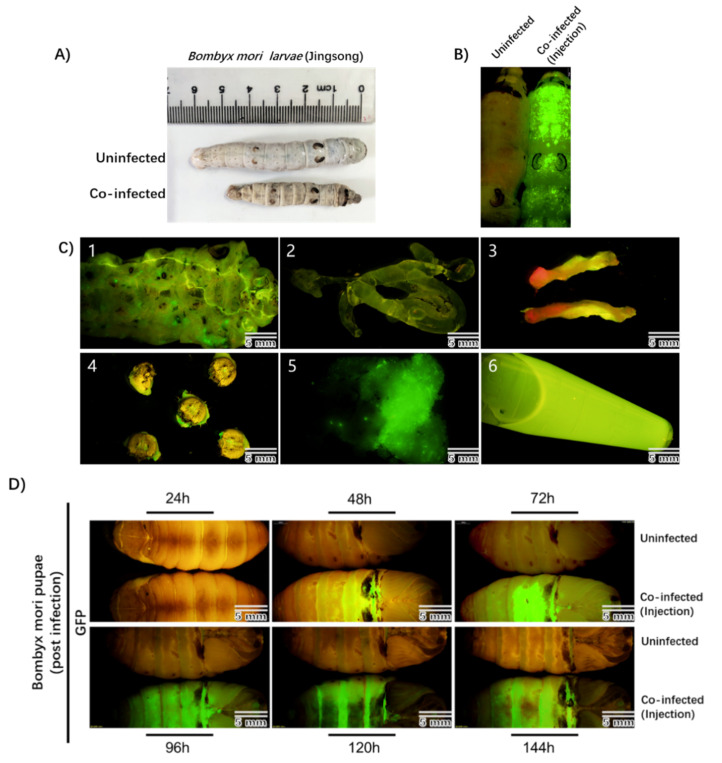
Observation of the *Bombyx mori* larvae and pupae after co-infection by rBmNPV/AAV2Rep-HBoV1Cap and rBmNPV/AAV2ITR-eGFP injection. (**A**) Morphology of uninfected and co-infected *Bombyx mori* larvae (Jingsong). (**B**) The images of larvae’s bodies obtained with fluorescence microscopy. The left sample is an uninfected larva, and the right sample is a co-infected larva. The image was taken 96 h after injection. (**C**) eGFP expression in tissues of *Bombyx mori* larvae. The larvae at 96 h of co-infection time were dissected to determine the eGFP expression in the cuticula (1), silk gland (2), midgut (3), cephalic region (4), fat body (5), and hemolymph (6). (**D**) The upper samples are uninfected pupae, and the lower, co-infected pupae. The images were taken at 24, 48, 72, 96, 120, and 144 h post-infection under a fluorescence microscope in complete darkness. The photographs were taken with an Olympus SZX16 fluorescence microscope at a low magnification. Scale bars, 5 mm.

**Figure 4 viruses-13-00704-f004:**
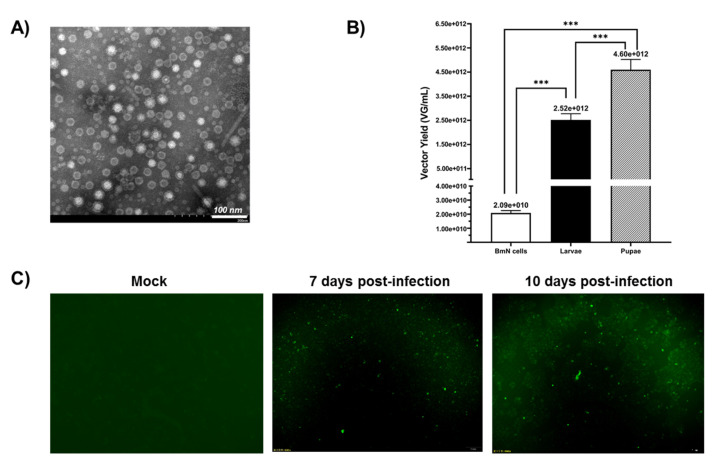
(**A**) Transmission electron microscopy analysis of negatively stained purified rAAV2/HBoV1. Original magnification × 100,000. Scale bars, 100 nm. (**B**) Analysis of the yields for rAAV2/HBoV1 derived from the BmN cells and Bombyx mori larvae and pupae. The yields of the rAAV2/HBoV1 vector were determined as vector genomes per mL (VG/mL). During sample processing, each larva or pupa was prepared in 1 mL of PBS, and 1 mL of cell suspension contained ~1 × 106 cells. A small segment of eGFP gene was amplified as the target gene detection. Six dilutions were de-tected for each sample, and the final viral titer was calculated by selecting samples with more standard Ct values. All experiments were done in triplicate. Mean ± SD values are presented. Asterisks depict Tukey’s multiple comparison test significance between groups following ANOVA, *** *p* < 0.001. (**C**) Transduction activity analysis of pupa-derived rAAV2/HBoV1 in 16HBE cells. 16HBE cells were transduced with rAAV2/HBoV1 at an MOI of ~ 2000 VG/cell and were examined for eGFP expression at 7- or 10-days post-transduction. Images were taken with an Olympus IX73 fluorescence microscope at a magnification of ×20.

## Data Availability

The data that support the findings of this study are available from the corresponding author upon reasonable request.

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
