# Peer review of "Bombyx mori Pupae Efficiently Produce Recombinant AAV2/HBoV1 Vectors with a Bombyx mori Nuclear Polyhedrosis Virus Expression System"

_viruses, 2021, doi:10.3390/v13040704_

Round 1
Reviewer 1 Report
The present paper deals with a very interesting topic, namely the use of B. mori as a bioreactor. The overall experiments and idea are clear to me but since this paper was submitted as a communication I would try to be more concise and explain some technical details in the supplementary materials instead of materials and methods, also because the vector construction is very similar in principle with other papers. In general, the language should be more concise and I would reduce the length of Results and Discussion sections.
Line 24: “has shown to be stable”
Line 25: “no apparent decrease in protein expression after five passages.” Is that correct or is it four passages as stated in lines 262? I perfectly understand it is not an on/off threshold, but I would doublecheck and try to be more coherent.
Line 46: RAAV2; I think it is serotype 2 but it is not stated anywhere
Line 88: “which was described”
Line 95: I would insert a reference about the meaning of codon-optimized Rep 78
Line 97: “Secondly, we also synthesized”
Line 106-109: brief explanation why you use TCID50 and pfu to titre the virus and not the same measure.
Line 111, Figure 1: “Scheme of the recombinant”; also, there is no explanation about the helper plasmid that is necessary for recombination neither in the text nor in the caption. Since it is drawn in the figure, I would say a few words about its function in the Bac-to Bac system.
Line 129: “they were injected with 5 ul” without by injection sounds better to me
Line 135: “and stored” only instead of “to collect and store”
Line 136 and in in general throughout the text, more concise if possible: “Five-six days after pupation, chrysalides were injected…”. Almost the same at line 140: 3-4 days post infection, pupae were stored at -80 °C until use.
Line 159: “The rAAV was purified by means of ultracentrifugation using two centrifugation media”
Line 192: why do you use tubulin AB? I guess for normalization. I would state this clearly.
Line 204-205: I would put the primer sequences in the supplementary materials and if you do, then I would insert the sequences of all the primers you use, not just the GFP primers.
Line 229: in a previous section there are details about the antibodies; I would then do the same for the GFP antibody.
Line 240-250: I am not sure there is a consistent use of MOI. A value of 0.5 for each vector is equal to a total MOI of 1 but I can’t understand the MOI value of 0.1.
Line 251: 72 h or 96 h post infection? It is 72 in the text but 96 in the caption of figure 2
Line 256: eGFP or GFP as elsewhere in the text?
In general, make a consistent use of “Rep78/Rep52” or “opt78Rep” etc.
Line 258-264 and Figure 2 caption: it is stated that Rep proteins expression between passage 1 and 4 is stable but it looks like there is a decrease, in lane P3 for instance, also because the tubulin expression looks the same to me. Can you explain this? Also, VP1 and VP2 are the Cap proteins I guess, but it is never stated clearly although they are generally mentioned as the proteins expressed by the HBoV1Cap. Can you be clearer on this?
Line 289-299: first, you inject the worms and then checked the exogenous protein expression, but it is not stated how. In the pupae you do the same and use the antibody; I guess it is the same in both conditions.
Line 303: it is stated that 5 pupae express the same protein amount than 1 L of insect cells. I wanted to see how this is determined but I was not able to find the method in the cited reference [14]; please check this.
Figure 3: I can’t see the scale bars
I perfectly understand it is not an easy task but when comparing the protein production of a cell line and a pupa/silkworm how do you normalize? A silkworm or a pupa have different sizes. You compared 1 ml of cell culture with one pupa/worm in 1 ml of PBS. Why did you choose one animal instead of, for instance, a certain amount of powder since the animals were frozen and grinded?
Figure 4: provide some details about the statistical analysis
Line 247: “The average yield of vectors per larva was”
Line 439: is or is not appropriate for transduction?
Author Response
Point 1:The present paper deals with a very interesting topic, namely the use of B. mori as a bioreactor. The overall experiments and idea are clear to me but since this paper was submitted as a communication, I would try to be more concise and explain some technical details in the supplementary materials instead of materials and methods, also because the vector construction is very similar in principle with other papers. In general, the language should be more concise and I would reduce the length of Results and Discussion sections.
Response 1: Considering that our paper is "Communication", we really should write it more concisely. Therefore, we reduced the content of materials and methods. Firstly, the details of two recombinant palsmids construction were showed in the supplementary materials. Secondly, in “rAAV Production and Purification”, the specific virus purification methods were transferred to supplementary materials.
Line 24: “has shown to be stable”
Response 2: Changed “was” → “has”
Line 25: “no apparent decrease in protein expression after five passages.” Is that correct or is it four passages as stated in lines 262? I perfectly understand it is not an on/off threshold, but I would doublecheck and try to be more coherent.
Response 3: Changed “The rBmNPV was shown to be stable and showed no apparent decrease in protein expression after five passages” → “The rBmNPV can express stably for at least five passages”.
Line 46: RAAV2; I think it is serotype 2 but it is not stated anywhere.
Response 4: Changed “rAAV2” → “rAAV serotype 2”.
Line 88: “which was described”
Response 5: Added “was”.
Line 95: I would insert a reference about the meaning of codon-optimized Rep 78
Response 6: Added reference: → Smith, R.H.; Levy, J.R.; Kotin, R.M. A simplified baculovirus-AAV expression vector system coupled with one-step affinity purification yields high-titer rAAV stocks from insect cells. Mol. Ther. 2009, 17, 1888–1896.
Line 97: “Secondly, we also synthesized”
Response 7: Changed “they” → “we”.
Line 106-109: brief explanation why you use TCID50 and pfu to titre the virus and not the same measure.
Response 8: The TCID50 method requires shorter detection time and easier operation than the PFU method, therefore our first choice is the TCID50 method for rBmNPV carried eGFP gene. The green fluorescence could be detected directly by Fluorescence microscope. The BmNPV/Rep-Cap does not carry the eGFP gene and can only be titrated using PFU.
Line 111: Figure 1: “Scheme of the recombinant”; also, there is no explanation about the helper plasmid that is necessary for recombination neither in the text nor in the caption. Since it is drawn in the figure, I would say a few words about its function in the Bac-to Bac system.
Response 9: Changed “Schematic” → “Scheme”. For the helper plasmid in this picture, the mini-Tn7 element on the pFastBac plasmid can transpose to the mini-attTn7 target site on the bacmid to construct the recombinant bacmid in the presence of transposition proteins provided by the helper plasmid. We added this explanation into the legend of Figure1.
Line 129: “they were injected with 5 ul” without by injection sounds better to me.
Response 10: Removed “by injection”.
Line 135: “and stored” only instead of “to collect and store”
Response 11: Changed “store” → “stored”.
Line 136: and in in general throughout the text, more concise if possible: “Five-six days after pupation, chrysalides were injected…”. Almost the same at line 140: 3-4 days post infection, pupae were stored at -80 °C until use.
Response 12: We've simplified our presentation:
In line 135, Removed “3-4 days”.
In line 137, Changed “after 5-6 days of pupae formation, they were...” → “pupae were...”.
In line 141, Removed “After 3-4 days post-infection,”
Line 159: “The rAAV was purified by means of ultracentrifugation using two centrifugation media”
Response 13: Changed “The method of rAAV purification was ultracentrifuged, in which we used two centrifugation media,” → “The rAAV was purified by means of ultracentrifugation using two centrifugation media”
Line 192: why do you use tubulin AB? I guess for normalization. I would state this clearly.
Response 14: Added “to normalize the added protein content”.
Line 204-205: I would put the primer sequences in the supplementary materials and if you do, then I would insert the sequences of all the primers you use, not just the GFP primers.
Response 15: We have added all the primers sequences of this experiment in the supplementary materials.
Line 229: in a previous section there are details about the antibodies; I would then do the same for the GFP antibody.
Response 16: The details about the eGFP antibody have been added in the materials and methods.
Line 240-250: I am not sure there is a consistent use of MOI. A value of 0.5 for each vector is equal to a total MOI of 1 but I can’t understand the MOI value of 0.1.
Response 17: Exactly, the recombinant BmNPV infected cells at the MOI of 0.5(total MOI of 1). We made a mistake at this point.
In line 252, “0.1” → “0.5”.
Line 251: 72 h or 96 h post infection? It is 72 in the text but 96 in the caption of figure 2
Response 18: For serial amplification passages from Passage 1 to Passage 8, viruses were collected at 72h after infection. For the test of passage stability, viruses were collected at 96h after co-infection. But in line 256, we indeed made a mistake in this point.
In line 256, “72” → “96”.
Line 256: eGFP or GFP as elsewhere in the text?
Response 19: In fact, the pFastBac-AAV2ITR-GFP plasmid contains the enhancer of GFP, so the whole article should be eGFP. This is also a clerical error. We have been revised all in the text.
In general, make a consistent use of “Rep78/Rep52” or “opt78Rep” etc.
Response 20: We have revised it and consisted as optRep78/Rep52.
Line 258-264 and Figure 2 caption: it is stated that Rep proteins expression between passage 1 and 4 is stable but it looks like there is a decrease, in lane P3 for instance, also because the tubulin expression looks the same to me. Can you explain this? Also, VP1 and VP2 are the Cap proteins I guess, but it is never stated clearly although they are generally mentioned as the proteins expressed by the HBoV1Cap. Can you be clearer on this?
Response 21: Thank you for point out this illogical WB result. We’ve thought that may cause by a confused collection of P2 and P3 sample during the preparation which we should explain in the original manuscript. And that is not rigorous enough to support our downstream experiments. We’ve repeated three times of this experiment later, and provided a revised fig in the new version. In fact, the P3 stock showed the highest protein expression level and the best stability.
Line 289-299: first, you inject the worms and then checked the exogenous protein expression, but it is not stated how. In the pupae you do the same and use the antibody; I guess it is the same in both conditions.
Response 22: We’ve added the WB results of AAV2 Rep and HBoV1 Cap expression after co-infection of the Bombyx mori larvae and pupae in the supplementary materials, as well as the eGFP and β-tubulin data.
Line 303: it is stated that 5 pupae express the same protein amount than 1 L of insect cells. I wanted to see how this is determined but I was not able to find the method in the cited reference [14]; please check this.
Response 23: We have added the correct reference. It is “Yin X,Li Z,Li J,et al..Rabies virus nucleoprotein expressed in silkworm pupae at high-levels and evaluation of immune responses in mice[J].J.Biotechnol.,2013,163( 3) : 333-338”.
Figure 3: I can’t see the scale bars.
Response 24: We’ve added a 5mm scale bar on each pic.
I perfectly understand it is not an easy task but when comparing the protein production of a cell line and a pupa/silkworm how do you normalize? A silkworm or a pupa have different sizes. You compared 1 ml of cell culture with one pupa/worm in 1 ml of PBS. Why did you choose one animal instead of, for instance, a certain amount of powder since the animals were frozen and grinded?
Response 25: It is difficult to normalize cell line and larvae/pupae for comparing the exogenous protein expression. This ‘1 ml normalization’ was designed for comparing the production of complete rAAV2/HBoV1 vectors (VG) between BmN cell line and silkworm larvae/pupae. We preserved one larva/pupa into 1ml PBS, and undergo the same procedure as 1ml infected BmN cell suspension (≈1x106 cells). Finally, we converted the complete vector production efficiency between pupae and cells based on qPCR results. We thought that may be a reasonable way to prove the high production efficiency of silkworm pupae in this work.
Figure 4: provide some details about the statistical analysis.
Response 26: We’ve added some details about the statistical analysis in the legend of Fig4.
Line 247: “The average yield of vectors per larva was”
Response 27: Changed “the average rAAV2 yield per larva was” → “the average yield of vectors per larva was”.
Line 439: is or is not appropriate for transduction?
Response 28: Should be ‘is not’. As a matter of fact, the rAAV2/HBoV1 only specifically infects human airway epithelial cells. Therefore, this experiment can only prove that rAAV2/HBoV1 has the transduction activity, but not whether it is appropriate for transduction. 16HBE is not the permissive cell line for the rAAV2/HBoV1 vectors. Since the vector has a preference to apical infection of polarized airway cells and expresses a high fluorescent signal, we will examine the vector transduction in polarized airway cells in the future.
Reviewer 2 Report
Review of Manuscript “Bombyx mori Pupae Efficiently Produce Recombinant AAV2/HBoV1 Vectors with a Bombyx mori Nuclear Polyhedrosis Virus Expression System” by Yu et al..
In their study the authors describe a new baculovirus-based packaging systems for chimeric rAAV2-based vectors pseudotyped with the human bocavirus type 1 capsid proteins (rAAV2/HBoV1). Bombyx mori pupae were used as a production platform after co-infection with two recombinant Bombyx mori nuclear polyhedrosis viruses (BmNPVs) providing the AAV2-ITR flanked EGFP transgene cassette and the AAV2-Rep/HboV1 Cap proteins, respectively. The authors found the yield of the rAAV2/HBoV1 vector in silkworm pupae to be higher than that in BmN cells and silkworm larvae and superior to that achieved in the AcMNPV/Sf9 system. Although the system has some potential as a rAAV2/HBoV1 bioreactor production system, at the moment the vectors still exhibit a low transduction activity, which may also be linked to the high percentage of empty virus particles observed.
Nevertheless it is an interesting system for future optimization and I would recommend publication, if the major and minor points listed in detail below were addressed in a revised version of the manuscript.
Detailed points of criticism:
Major points:
1) Fig. 2, infection of BmN cells with different passages of recombinant BmNPV stocks: Whereas the EGFP expression as judged by fluorescence microscopy is highest in passage 3 (P3), the AAV2 Rep, HBoV1 Cap and also the EGFP levels in P3 seem to be clearly lower than in the passages P2 and P4, when assayed by western blot. This does not seem to be due to protein loading, since the loading control ß-tubulin looks quite constant in P2 to P4. Since the P3 stocks were used for the subsequent rAAV2/HBoV1 packaging experiments, the authors should comment on this.
2) The expression data of AAV2 Reo and HBoV1 Cap proteins after co-infection of the Bombyx mori larvae (lines 293 to 295) and pupae (lines 300 to 301) should be presented for better evaluation.
3) The transduction rates for the rAAV2/HBoV1-EGFP vectors shown in fig. 4C seem to be very low, even 10 days post transduction. Nevertheless, the percentage of GFP-positive HBE16 cells should be determined by FACS analysis and represented. The corresponding values from transductions with vectors produced in larvae and BmN cells should ideally also be included for better comparison.
Minor points:
1) Line 88: Typo, either phrase “, which was described“ or “cloning vector described“ (omit “which“).
2) Line 97: Typo, phrase “we synthesized“ or “a fragment was synthesized“.
3) Line 133: Omit the word “after“ (also in line 146)
4) Line 136: Replace “they“ by “these“.
5) Material and Methods section, line 142: I would suggest describing the generation of the recombinant BmNPV stocks in a section separate from the rAAV2/HBoV1 vector production.
6) Line 142: I would suggest to replace “experiments“ by the term “purification steps“.
7) Line 159: Use the term “ultracentrifugation“ instead of “ultracentrifuged“.
8) For clarity and consistence, please generally refer to the produced vectors as rAAV2/HboV1 vectors and not as rAAV vectors as sometimes done in the manuscript. Furthermore, refer to the rBmNPV used for production as BmNPV/AAV2Rep-HBoV1Cap and BmNPV/AAV2ITR-GFP consistently (and not as in line 238).
Author Response
In their study the authors describe a new baculovirus-based packaging system for chimeric rAAV2-based vectors pseudotyped with the human bocavirus type 1 capsid proteins (rAAV2/HBoV1). Bombyx mori pupae were used as a production platform after co-infection with two recombinant Bombyx mori nuclear polyhedrosis viruses (BmNPVs) providing the AAV2-ITR flanked EGFP transgene cassette and the AAV2-Rep/HboV1 Cap proteins, respectively. The authors found the yield of the rAAV2/HBoV1 vector in silkworm pupae to be higher than that in BmN cells and silkworm larvae and superior to that achieved in the AcMNPV/Sf9 system. Although the system has some potential as a rAAV2/HBoV1 bioreactor production system, at the moment the vectors still exhibit a low transduction activity, which may also be linked to the high percentage of empty virus particles observed.
Nevertheless it is an interesting system for future optimization and I would recommend publication, if the major and minor points listed in detail below were addressed in a revised version of the manuscript.
Detailed points of criticism:
Major points:
1) Fig. 2, infection of BmN cells with different passages of recombinant BmNPV stocks: Whereas the EGFP expression as judged by fluorescence microscopy is highest in passage 3 (P3), the AAV2 Rep, HBoV1 Cap and also the EGFP levels in P3 seem to be clearly lower than in the passages P2 and P4, when assayed by western blot. This does not seem to be due to protein loading, since the loading control ß-tubulin looks quite constant in P2 to P4. Since the P3 stocks were used for the subsequent rAAV2/HBoV1 packaging experiments, the authors should comment on this.
Response 1: Thank you for point out this illogical WB result. We’ve thought that may cause by a confused collection of P2 and P3 sample during the preparation which we should explain in the original manuscript. And that is not rigorous enough to support our downstream experiments. We’ve repeated three times of this experiment later, and provided a revised fig in the new version. In fact, the P3 stock showed the highest protein expression level and the best stability.
2) The expression data of AAV2 Rep and HBoV1 Cap proteins after co-infection of the Bombyx mori larvae (lines 293 to 295) and pupae (lines 300 to 301) should be presented for better evaluation.
Response 2: We’ve added the WB results of AAV2 Rep and HBoV1 Cap expression after co-infection of the Bombyx mori larvae and pupae in the supplementary materials. Together with the eGFP and β-tubulin data.
3) The transduction rates for the rAAV2/HBoV1-EGFP vectors shown in fig. 4C seem to be very low, even 10 days post transduction. Nevertheless, the percentage of GFP-positive HBE16 cells should be determined by FACS analysis and represented. The corresponding values from transductions with vectors produced in larvae and BmN cells should ideally also be included for better comparison.
Response 3: 16HBE is not the permissive cell line for the rAAV2/HBoV1 vectors. Since the vector has a preference to apical infection of polarized airway cells and expresses a high fluorescent signal (Ref: 25. Deng, X.; Zou, W.; Yan, Z.; Qiu, J. Establishment of a recombinant aav2/hbov1 vector production system in insect cells. Genes 2020, 11, 439.), we will examine the vector transduction in polarized airway cells using FACS in the future. Actually, we’ve tried different ways to obtain the HAE-ALI cell line (from local hospital or import from aboard), but it is a bit difficult to us at this moment due to the COVID-19 pandemic. We are still working on that, and also trying to optimize this ‘pupae expression system’. We will improve these data in the future.
Minor points:
- Line 88: Typo, either phrase ‘, which was described’ or ‘cloning vector described’ (omit ‘which’).
Response 4: Changed ‘which described’ à ‘which was described’.
- Line 97: Typo, phrase ‘we synthesized’ or ‘a fragment was synthesized’.
Response 5: Changed ‘they also synthesized a fragment’ à ‘we synthesized’.
- Line 133: Omit the word ‘after’(also in line 146)
Response 6: Omitted ‘after’ both in line 133 and line 146.
- Line 136: Replace “they “by “these “.
Response 7: Changed ‘they’ à ’these’.
- Material and Methods section, line 142: I would suggest describing the generation of the recombinant BmNPV stocks in a section separate from the rAAV2/HBoV1 vector production.
Response 8: We’ve retained the describing of BmNPV stocks part in this section, and removed the vector purification steps to the supplementary materials.
- Line 142: I would suggest to replace “experiments“ by the term “purification steps“.
Response 9: Changed ‘experiments’ à ‘purification steps’.
- Line 159: Use the term “ultracentrifugation “instead of “ultracentrifuged “.
Response 10: Changed ‘ultracentrifuged’ à ‘ultracentrifugation’.
8) For clarity and consistence, please generally refer to the produced vectors as rAAV2/HboV1 vectors and not as rAAV vectors as sometimes done in the manuscript. Furthermore, refer to the rBmNPV used for production as BmNPV/AAV2Rep-HBoV1Cap and BmNPV/AAV2ITR-GFP consistently (and not as in line 238).
Response 11: We’ve consisted the produced vectors as rAAV2/HBoV1 in the revised version. We’ve also consisted all the rBmNPV as rBmNPV/AAV2Rep-HBoV1Cap or rBmNPV/AAV2ITR-eGFP on the revised version.
Round 2
Reviewer 1 Report
Dear Sirs, thanks for considering my advices. I have just one more very little advice about figure 4 caption to complete the details about the statistical analysis. I would also add which type of analysis you carried out (I guess ANOVA + post-hoc; for the latter which one) and personally, as matter of transparency, I would add the degrees of freedom and the p-value of the ANOVA like you did for the p-value for the post-hoc test. Apart from that, the paper is much clearer and more in compliance with a “communication”.
Author Response
Thank you so much for this meticulous and professional advice. For the details about the statistical analysis figure 4B, we’ve used another software Graph Pad to draw the figure, and at the same time, variance analysis and analysis of P value were carried out. Asterisks depict Tukey’s multiple comparison test significance between groups following ANOVA. In addition, in order to make the yield of BmN cells in the figure more prominent, we also processed the Y-axis coordinates. We’ve changed a more detailed graph for Fig4B, and also improved the legend of that.
Reviewer 2 Report
Review of revised version of manuscript “Bombyx mori Pupae Efficiently Produce Recombinant AAV2/HBoV1 Vectors with a Bombyx mori Nuclear Polyhedrosis Virus Expression System” by Yu et al..
In the revised version, most of the major and minor points raised in my review of the original manuscript have been addressed nicely. The reasons the authors state for not being able to obtain the HAE-ALI cell line for monitoring transduction of the purified rAAV2/HBoV1 vectors are comprehensible. I would therefore like to recommend accepting the paper in its present form.
Author Response
Thank you so much for all of the meticulous and professional advices.